# Ambient 🛡 Proteins
# Training Diffusion Models on Noisy Structures

## Abstract

We present *Ambient Protein Diffusion*, a framework for training protein diffusion models that generates structures with unprecedented diversity and quality. State-of-the-art generative models are trained on computationally derived structures from AlphaFold2 (AF), as experimentally determined structures are relatively scarce. The resulting models are therefore limited by the quality of synthetic datasets. Since the accuracy of AF predictions degrades with increasing protein length and complexity, de novo generation of long, complex proteins remains challenging. Ambient Protein Diffusion overcomes this problem by treating low-confidence AF structures as corrupted data. Rather than simply filtering out low-quality AF structures, our method adjusts the diffusion objective for each structure based on its corruption level, allowing the model to learn from both high and low quality structures. Empirically, ambient protein diffusion yields major improvements: on proteins with 700 residues, diversity increases from 45% to 85% from the previous state-of-the-art, and designability improves from 70% to 88%.

## 1 Introduction

Proteins are the fundamental building block of life. They accelerate chemical reactions by many orders of magnitude, convert sunlight into food, and underpin the myriads of processes within cells and organisms with the level of accuracy and precision required to sustain life [6, 25]. Unlike computational protein engineering—which focuses on improving the developability or function of existing proteins through computationally guided mutations for practical biotechnological applications [17, 31, 20, 32, 8, 29, 16]—*de novo* protein design aims to create entirely new proteins with specified structures and functions, ultimately seeking to discover folds and activities not found in nature [10]. Since protein function is largely determined by tertiary and quaternary structure, generative machine learning frameworks for protein design focus on learning the sparse, evolutionarily sampled landscape of protein structures, with the goal of generating novel, functional backbone scaffolds beyond those observed in nature [40, 26, 27, 23, 19, 42, 18, 7, 44, 37].

Recent breakthroughs in machine learning–based structure prediction—most notably AlphaFold2 [24]—have made it possible to infer accurate protein structures directly from sequence [24, 9, 28]. This progress has enabled the creation of large-scale structural resources such as the AlphaFold Database (AFDB), which contains over 214M predicted structures from UniProtKB sequences [11, 36]. In parallel, high-throughput tools for sequence and structure comparison, such as MMSeqs2 and FoldSeek, have facilitated the curation of large, diverse training datasets from AFDB [5]. Among them, the 2.3M AFDB cluster dataset, has already been shown to improve the capabilities of generative models for protein structure design [27, 19].

The quality of a generative model depends on the size and fidelity of its training data. While AlphaFold2 (AF) has enabled large-scale protein structure prediction, its outputs often contain biological or computational inaccuracies [41]. To estimate the reliability of a predicted structure, AlphaFold provides a per-residue confidence score, the predicted Local Distance Difference Test (pLDDT), which is a proxy of local structural accuracy. In practice, researchers frequently filter predicted structures based on average pLDDT scores, training only on high-confidence subsets (typically using a cutoff of pLDDT $> 80$). Lower pLDDT scores are disproportionately associated with longer and more structurally complex proteins. As a result, filtering based on pLDDT introduces a bias toward smaller, simpler folds, reducing structural diversity in the training set and impairing the model's ability to generalize to more complex regions of structure space—including longer proteins. Notably, many low-pLDDT structures still contain well-folded domains that are misoriented with respect to each other, as reflected by low predicted alignment error (pAE). These structures can still offer valuable domain-level and coarse-grained information about the structure distribution, which is discarded by overly aggressive filtering.

To mitigate these issues, we depart from the standard paradigm of filtering low-confidence structures. Instead, we introduce *Ambient Protein Diffusion* —a framework for training diffusion models that incorporates proteins with noisy or incomplete structures directly into the training process. Ambient Protein Diffusion builds on recent advances in learning generative models from corrupted data [12, 14, 1, 13, 2, 33, 4, 39, 15, 30, 34], which have explored controlled corruption settings such as additive Gaussian noise [12, 14, 1, 15] and masking [13, 2]. Our framework generalizes these techniques to arbitrary, unknown corruption processes, enabling the training of generative models in scientific domains where the corruption mechanism is complex and non-parametric. In our setting, the AlphaFold prediction errors represent such a corruption: they are structured, not explicitly modeled, and vary across protein size and topology. Yet, our method effectively leverages these imperfect samples to significantly advance the capabilities of generative protein models. For example, on proteins with 700 residues, our 16.7M parameter model our method improves diversity from 45% to 85% and increases designability from 70% to 88% compared to the previous state-of-the-art, Proteína, a 200M parameter model. Below, we summarize our key contributions:

- We generalize recent approaches for training generative models on corrupted data to handle arbitrary, non-parametric, and unknown corruption processes, enabling their application to scientific domains.

- We construct a new training set from the AFDB cluster dataset optimized for geometric diversity, rather than evolutionary similarity, yielding a broader and more representative sampling of structural space for generative modeling.

- We demonstrate that Ambient Protein Diffusion effectively leverages low-pLDDT AlphaFold predictions, allowing the model to learn from all available samples without distorting the underlying structure distribution.

- We achieve state-of-the-art results in both diversity and designability for protein generation, improve diversity by 45% and designability by 18% on long proteins (up to 800 residues), and establish the Pareto frontier between these objectives on short proteins ($< 256$ residues).

## 2 Background and Related Work

**De novo Protein Generation.** Most *de novo* protein generation frameworks that operate in structure space follow a three-step pipeline: (1) a generative model samples a three-dimensional backbone structure; (2) an inverse folding model (e.g., ProteinMPNN) proposes amino acid sequences likely to fold into the generated backbone; and (3) these sequences are evaluated by a structure prediction model (e.g., ESMFold) to identify the ones that best recapitulate the target fold.

Pioneering methods such as RFDiffusion [40] and Chroma [23] have established strong baselines for backbone generation. More recent advances include Genie [26], which introduces a denoising diffusion model with an SE(3)-equivariant network that generates proteins as point clouds of reference frames; Genie2 [27], which scales Genie using synthetic AlphaFold structures to improve training data diversity; and Proteína [19], which replaces diffusion with flow matching and scales both model size and dataset scale by orders of magnitude to improve performance on longer and more complex monomeric proteins.

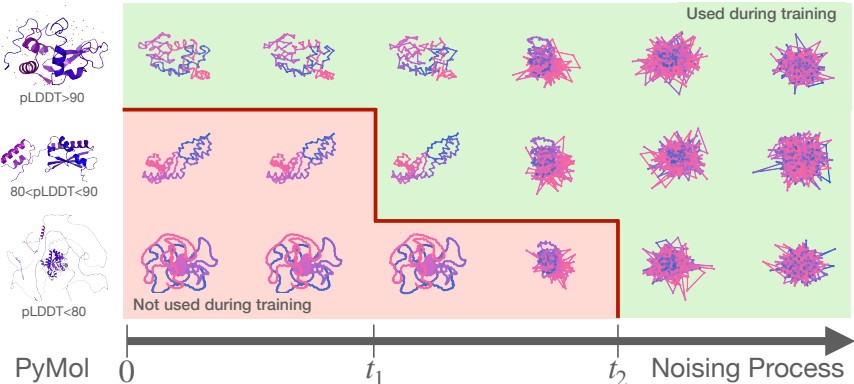

Figure 1: **Overview of Ambient Protein Diffusion on alphafold structures.** Rows 1-3 show the noising process (from left to the right) of three different alphafold proteins based on their average pLDDT (top: high, middle: medium, and bottom:low). These proteins are only used during training at the green diffusion times. At these noise levels, any initial AlphaFold prediction errors in low-pLDDT proteins have effectively been "erased" by the added noise, and the distributions of low- and high-pLDDT proteins have merged.

Ambient Protein Diffusion is built using the Genie architecture and makes use of ambient protein diffusion to achieve state-of-the-art results with substantially shorter training times, much fewer parameters (16.7M vs 200M), and significantly reduced computational requirements.

**Training Datasets.** Recent advances in structure prediction—most notably AlphaFold2 [24] and ESMFold [28]—have dramatically expanded the available structural data, enabling the prediction of ∼214M and ∼617M monomeric protein structures from UniProtKB (via the AlphaFold Protein Structure Database) [36] and metagenomic libraries (via the ESM Atlas [28], respectively. While this explosion of computational structures presents unprecedented opportunities, it also poses significant challenges for downstream bioinformatic analysis and model training, particularly due to the scale, redundancy, and uneven quality of the predicted structures. To address this, prior work applied MMSeqs2 [22] and FoldSeek [35] to cluster the AlphaFold Database (AFDB), yielding ∼2.3M clusters designed to capture evolutionary relationships between predicted structures [5]. This AFDB cluster dataset has since served as the foundational dataset to train several generative protein structure models [19, 27].

In this work, we revisit the FoldSeek pipeline applied AFDB with a different objective: rather than optimizing for evolutionary insight, we recluster the AFDB cluster dataset with hyperparameters tuned to maximize geometric diversity. Our goal is to construct a dataset better suited for learning a generative model of structural space—one that emphasizes structural rather than evolutionary variation. Starting from the 2.3 million AFDB clusters, we use the cluster representatives with average pLDDT > 70 (∼1.29M structures) and apply our geometric clustering procedure. The resulting dataset comprises roughly ∼292K structurally diverse clusters.

**Diffusion Models.** The goal in diffusion modeling is to sample from an unknown density $p_0$ that we have sample access to. Formally, let $\mathcal{D} = \{x_0^i\}_{i=1}^N$ a dataset of $N$ independent samples, where $X_0^i \sim p_0$. The unknown distribution $p_0$ is potentially complex, high-dimensional and multimodal. To make the sampling problem more tractable, in diffusion modeling we target smoothened densities $p_t$ defined as the convolution with a Gaussian: $p_t = p_0 * \mathcal{N}(0, \sigma(t)I_d)$[1], where $\sigma(t)$ is an increasing function of $t$, with $\sigma(0) = 0$. In particular, the object of interest in diffusion modeling is the score-function of the smoothened densities, defined as $\nabla_{x_t} \log p_t(x_t)$. The latter is connected to the optimal denoiser (in the $l_2$ sense) through Tweedie's Formula: $\nabla \log p_t(x_t) = \frac{\mathbb{E}[X_0|X_t=x_t]-x_t}{\sigma^2(t)}$.

---

[1]Alternative formulations of diffusion modeling, such as the Variance Preserving case, are equivalent to this case up to a simple reparametrization. For the ease of analysis, we focus our presentation on corruptions of the form $X_t = X_0 + \sigma_t Z, \quad Z \sim \mathcal{N}(0, I_d)$.

Given access to $\mathbb{E}[X_0|X_t = x_t]$ one can sample from the distribution $p_0$ of interest by running a reverse diffusion process [3]. Hence, the sampling problem becomes equivalent to the problem of approximating the set of functions $\{\mathbb{E}[X_0|X_t = \cdot]\}_{t=0}^T$. Given a sufficiently rich family of functions $\{h_\theta : \theta \in \Theta\}$, the conditional expectation at a particular time $t$ can be learned by minimizing the objective:

$$J(\theta) = \mathbb{E}_{t \in \mathcal{U}[0,T]}\mathbb{E}_{x_0,x_t|t}\left[||h_\theta(x_t) - x_0||^2\right]. \tag{1}$$

**Protein Diffusion.** In protein diffusion models that target backbone generation, $X_0$ captures the 3-D co-ordinates for each one of the backbone residues of the protein. The length of a protein varies and the standard practice is to pad each protein to specified length (256 for Genie [26, 27]), with some special mask indicating the valid positions.

**Learning from noisy data.** Recent work has explored the problem of learning diffusion models from corrupted data. Typically, the corruption process is simple, e.g. it can be additive Gaussian noise as in [12, 14, 1], or masking as in [1, 2]. Even in works that the corruption process is more general, the degradation needs to be known and multiple diffusion trainings are required until an Expectation-Maximization algorithm converges [33, 4]. In this work, we deviate from this setting as the corruption process is unknown and complex, which may include AlphaFold learning and hallucination errors, and noise inherent to the structural biology technique used to solve the structure, etc. We also target a single diffusion training instead of performing multiple EM iterations. The method is detailed in Section 3.

Our work generalizes the techniques developed in [12, 14] for the additive Gaussian noise case. Particularly, in [14], the authors consider learning from a dataset $\mathcal{D} = \{(x_{t_i}^i, t_i)\}_{i=1}^N$ of samples noised with additive Gaussian noise of different variances $\{\sigma^2(t_i)\}_{i=1}^N$. Formally, let $X_{t_i} = X_0 + \sigma(t_i)Z$, where $X_0 \sim p_0, Z \sim \mathcal{N}(0, I_d)$. Each point $X_{t_i}$ contributes to the learning only for $t \geq t_i$, using the objective:

$$\hat{J}(\theta) = \mathbb{E}_{t \in \mathcal{U}[0,T]} \sum_{x_{t_i} \in \mathcal{D}:\ t_i > t} \mathbb{E}_{x_t|x_{t_i},t_i}\left[||\alpha(t, t_i)h_\theta(x_t, t) + (1 - \alpha(t, t_i))x_t - x_{t_i}||^2\right], \tag{2}$$

$\alpha(t, t_i) = \frac{\sigma^2(t) - \sigma^2(t_i)}{\sigma^2(t)}$. As the number of samples grows to infinity, Equation 2 also recovers the conditional expectation $\mathbb{E}[X_0|X_t = x_t]$, but it does so while being able to utilize noisy samples. This objective recovers the true minimizer because one can prove that the conditional expectation $\mathbb{E}[X_{t_i}|X_t = x_t]$, lies in the line that connects the current noisy point $x_t$ and the prediction of the clean image, $\mathbb{E}[X_0|X_t = x_t]$.

## 3 Method

Formally, we are given access to samples from the AlphaFold distribution $\tilde{p}_0$ and aim to learn how to sample from the true distribution of experimentally solved structures, $p_0$, without an explicit degradation model mapping $p_0 \to \tilde{p}_0$. In the protein structure setting, it is not appropriate to model the structural deviations introduced by AlphaFold as additive Gaussian noise. Our key insight is that, regardless of how $\tilde{p}_0$ deviates from $p_0$, adding noise to both distributions causes them to contract toward one another. As the noise level increases, the distributions $\tilde{p}_t$ and $p_t$ become progressively more aligned. This is because it is known that Gaussian noise contracts distribution distances (KL divergence) in the following sense:

$$D_{KL}(p_t||\tilde{p}_t) \leq D_{KL}(p_{t'} || \tilde{p}_{t'}), \quad \forall t \geq t'. \tag{3}$$

In fact, as $t \to \infty$, we have that: $D_{KL}(p_t || \tilde{p}_t) \to 0$, as both distributions converge to the same Gaussian. We now define the concept of merging of two distributions towards the same measure.

**Definition 3.1 ($\epsilon$-merged)** *We say that two distributions, $p$ and $\tilde{p}$ are $\epsilon$-merged, if the KL distance between the two is upper-bounded, by $\epsilon$, i.e., if $D_{KL}(p || \tilde{p}) \leq \epsilon$.*

Similarly, we define the merging time of two distributions as the minimal amount of noise we need to add such that the two distributions become $\epsilon$-merged. Formally,

**Definition 3.2 ($\epsilon$-merging time)** *Let two distributions $p, \tilde{p}$. We define their $\epsilon$-merging time as follows:*
$$t_n(p, \tilde{p}, \epsilon) = \inf\{t : \mathrm{D_{KL}}(p * \mathcal{N}(0, \sigma(t)^2 I) \,||\, \tilde{p} * \mathcal{N}(0, \sigma(t)^2 I) \leq \epsilon\}.$$

Assuming we can estimate the $\epsilon$-merging time between two distributions $p$ and $\tilde{p}$, our key idea is to treat samples from $\tilde{p}_t$ as approximate samples from $p_t$ for all timesteps $t > t_n(p, \tilde{p}, \epsilon)$. This idea is illustrated in Figure 1. The intuition is that once the distributions have sufficiently merged under noise, the residual shift becomes negligible and samples from $\tilde{p}_t$ can be used for learning $p_t$. This holds because: (i) the learning algorithm may not be sensitive to small distributional discrepancies at high noise levels, and (ii) even if some bias is introduced, the remaining diffusion trajectory for times $t \leq t_n(p, \tilde{p}, \epsilon)$ is robust to small initial distributional mismatch due to its inherent stochasticity.

**Sample dependent noise levels.** At a high level, our objective is to determine the $\epsilon$-merging time between the distribution of AlphaFold-predicted structures and that of experimentally resolved proteins. A key challenge arises from the fact that the AlphaFold distribution is highly heterogeneous in structural fidelity—that is, the accuracy with which AlphaFold predicts the true protein structure varies widely across samples. It is well established that short, structurally simple proteins are predicted with higher confidence, while longer and more complex proteins tend to yield lower-confidence predictions. This trend is illustrated in Figure 2B (Left). If we were to assign a single noise level across the entire AlphaFold dataset, we would need to select a relatively high noise level to accommodate the lowest-confidence predictions, particularly from long proteins. This would unnecessarily degrade the training signal for high-confidence structures—regardless of protein length—and limit the model's ability to learn from clean supervision. To address this, we treat the AlphaFold dataset as a mixture of $K$ sub-distributions, $q_1, q_2, \ldots, q_K$, each representing a distinct confidence regime. We then assign each sub-distribution an appropriate noise level, sufficient to bring it $\epsilon$-close to the distribution of high-confidence structures under the same noise schedule. This formulation allows the model to effectively learn from high-confidence AlphaFold predictions and incorporate low-confidence structures in a controlled manner, mitigating the degradation typically caused by noisy training data.

A natural way to decompose the AlphaFold distribution into a mixture of quality-specific sub-distributions is to leverage AlphaFold's self-reported confidence metric—the average predicted Local Distance Difference Test (pLDDT) score—as a proxy for predicted structural fidelity. In particular, given a dataset $\mathcal{D} = \{(x_0^{(i)}, \mathrm{pLDDT}^{(i)})\}_{i=1}^N$, we consider $K$ distributions (where $K$ is a hyperparameter to be chosen) with empirical observations for the $j$-th distribution being all the samples $\{(x_0^{(i)}, \mathrm{pLDDT}^{(i)}) : c_{\min}^{(j)} \leq \mathrm{pLDDT}^{(i)} \leq c_{\max}^{(j)}\}$, for some hyperparameters $c_{\min}^{(j)}, c_{\max}^{(j)}$.

**Choice of sub-distribution boundaries.** In this work, we adopt a deliberately simple and conservative strategy by partitioning the AlphaFold dataset into three discrete quality regimes based on the average pLDDT score: high-quality proteins (pLDDT $> 90$), medium-quality proteins (pLDDT in $[80, 90]$) and low-quality proteins (pLDDT in $[70, 80]$). We acknowledge that this discretization is coarse and that more principled alternatives may yield further improvements—for instance, by optimizing the bin boundaries or learning a continuous mapping from pLDDT to diffusion time. Despite the simplicity of our choices, our experimental results demonstrate that even a naive quality-aware decomposition can lead to important gains in performance across both short and long proteins. There are two sources of benefit over filtering methdos: 1) low-quality data (previously discarded) give diversity and 2) the distinction we do between medium-quality and high-quality data increases designability.

**Ambient Protein Diffusion Algorithm.** Our algorithm takes as input a dataset of protein structures together with their average pLDDT score, $\mathcal{D} = \{(x_0^{(i)}, \mathrm{pLDDT}^{(i)})\}_{i=1}^N$, a diffusion schedule, $\sigma(t)$, and a mapping function $f : [0, 100] \mapsto \mathbb{R}^+$ that translates the average pLDDT value of a protein to its estimated $\epsilon$-merging time.

**Annotation stage.** The first step of the algorithm replaces each protein in the dataset with a noisy version of itself, where the noise level is determined by mapping function $f$. After this transformation, each protein can be treated as a sample from the target distribution convolved with a Gaussian at its assigned noise level. Importantly, this corruption step is only performed once during dataset preprocessing.

**Loss function.** After the annotation stage, we need to solve a training problem where we have data corrupted at different noise levels with additive Gaussian noise, as in [12, 14]. Hence, we can use the objective of Equation 2.

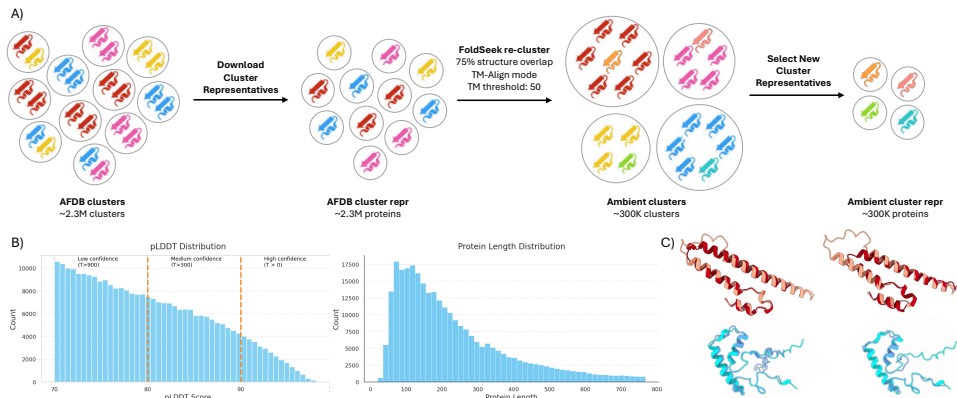

Figure 2: **Reclustering the AFDB cluster dataset for generative protein modeling.** (A) Starting from the 2.3M clusters in the AFDB, we cluster the representatives with FoldSeek with alignment-type set to TM-Align, TM threshold set to 0.5, and coverage set to 0.75. This results in 300K cluster (pLDDT > 70) from which we keep the representatives for training. (B) pLDDT and protein length statistics for our new training set. (C) Example of two protein clusters where two cluster members (red and blue) are superimposed with their cluster representative (beige and cyan).

Instead of directly applying the loss, we first need to rescale each time $t$ to account for the vanishing gradient effect that is due to the multiplicative factor $a(t)$. Specifically, we need to rescale the loss at time $t$ with: $w(t) = \frac{1}{a^2(t)} = \frac{\sigma^4(t)}{(\sigma^2(t)-\sigma^2(t_i))^2}$. We underline that this rescaling was not mentioned in the original paper of Daras et al. [14, 12], for training with noisy data. Yet, we find this rescaling critical for the success of our method. We hypothesize that the authors of [12, 14] did not encounter this issue because there were at most two noise levels considered, while in AF predicted protein structures there is a whole spectrum of assigned noise levels based on the predicted quality (measured by average pLDDT) of a protein structure. We provide further details about the loss implementation in the supplement and pseudocode for our Algorithm in the Appendix.

**Uniform Protein Sampling in terms of diffusion times.** To perform a training update for a diffusion model, we typically sample a point from the training distribution and then we uniformly sample the noise level $t$. However, since in our case we are dealing with noisy data, not all times $t$ are allowed for a given protein, i.e. a protein with $\text{pLDDT}^{(i)}$ is only used for times $t \geq f(\text{pLDDT}^{(i)})$. To avoid spending most of the training updates on very noisy proteins, we opt for sampling first the diffusion time and then select from the eligible proteins that can be used in that diffusion time. This strategy ensures balanced coverage across the diffusion trajectory—from low to high noise—while still leveraging the diversity of low-confidence structures (pLDDT < 80) in our training dataset.

**Reclustering AFDB clusters for generative modeling applications.** The AFDB cluster dataset [5] has been used to train several generative protein models [27, 19]. However, the original intent behind the clustering was to study structure evolution across AFDB. Thus, the hyperparameters were chosen to obtain clusters of homologous structures, and the authors report that 97.4% of pairwise comparisons within clusters are conserved at the H-group (Homology) level of the ECOD hierarchical domain classification (median TM-score 0.71). While these FoldSeek hyperparameters are well-suited for evolutionary analysis of AFDB, we found that the AFDB cluster dataset has a significant degree of structural duplication and near-duplication between clusters that are more distantly evolutionarily related (see Figure 2C). This structural redundancy leads to an imbalanced training set, where structural motifs from the larger protein superfamilies are overrepresented.

Given this finding, we hypothesized that the datasets for generative modeling of protein structures—particularly for backbone-based models— benefit more from clusters defined purely by geometric similarity. To address this, we constructed a new clustering dataset derived from the AFDB cluster representatives, with an exclusive focus on structural topology. Specifically, these are the changes we made to the FoldSeek hyperparameters: switched the alignment-type from 3Di+AA to TM-Align to improve fidelity, used a TM-score threshold of 0.5, and relaxed the alignment coverage

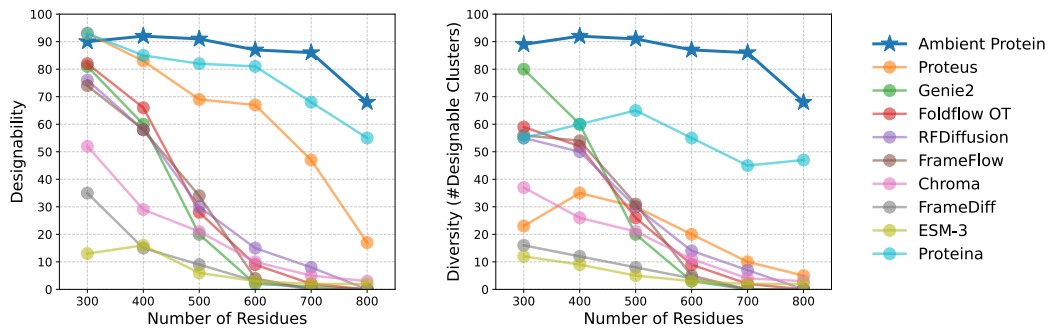

Figure 3: **Long protein generation performance.** We fine-tune Ambient Protein Diffusion on proteins up to 768 residues and sample sequences ranging from 300 to 800 residues. Ambient Protein Diffusion generates diverse and designable structures across this length range with consistent performance.

from 0.9 to 0.75. We relaxed the alignment coverage to improve clustering of AlphaFolded proteins with extended, unfolded N- or C-terminal regions (i.e., noodle tails) (Figure 2C). This approach produced a more balanced dataset that samples structural folds more uniformly, independent of their evolutionary relationships. Ablations showing the contribution of this reclustering versus our ambient training approach are given in Figure 4.

## 4   Experimental Results

We build on the Genie2 codebase [27]. Our model architecture follows the Genie2 architecture except that it is scaled larger, using 8 triangle layers as opposed to 5. We follow Genie2's schedule for inference time sampling and diffuse for 1000 timesteps.

We train Ambient Protein Diffusion in 3 stages with increasingly longer proteins. In first stage, we train on proteins from 50 to 256 residues for 200 epochs on our ambient clusters dataset using the representatives ($\sim$ 196,000 proteins). Since we increased the batch size to 384 items, we adopted a learning rate schedule to improve convergence [21]. We train with the AdamW optimizer with a maximal learning rate of $1.0 \times 10^{-4}$. During the second and third stage, we include additional cluster representatives of at most 512 and 712 residues, which scales our dataset to $\sim$269,000 and $\sim$291,000 proteins respectively. Training is performed on 48 GH200 GPUs and runs in 18, 48, and 48 hours for each stage respectively. More hyperparameters and details about our evaluation metrics can be found in the supplement. We underline that the computational cost of training our model, while significant, is still relatively low compared to the Proteína's estimated 14 days training on 128 A100 GPUs. This is due to the decreased size of our model ($<$ 17M vs 200M) and training set ($\sim$ 290K vs $\sim$ 780K). We further note that our goal is to develop models that perform well across a range of tasks, including long-protein generation, motif scaffolding, and more. To this end, we train only two models for the purposes of this paper: one model optimized for long-protein generation (Figure 3) and another optimized for short-protein generation (Figure 4).

**Comparisons on unconditional generation of longer proteins.**

In Figure 3, we compare Ambient Protein Diffusion performance on generating backbone for proteins with length ranging from 300 to 800 residues. To directly compare with Proteína on long-protein generation, we adopt its three-stage training and evaluation protocol. During training, the maximum sequence length is capped at 768 residues. For evaluation, we sample 100 protein backbones at each target length and evaluate them using the designability and diversity metrics. Since Ambient Protein Diffusion builds on Genie2, we use the same sampling procedure—running 1000 diffusion steps with a noise scale of $\gamma = 0.6$.

Ambient Protein Diffusion achieves designability and diversity scores exceeding 90% for proteins between 300 and 500 residues, and maintains scores above 85% for lengths up to 700 residues. For 800-residue proteins, both metrics decline to 68%. Compared to Proteína, Ambient Protein Diffusion outperforms by 26% in designability and 91% in diversity at length 700, and by 25% and 44%, respectively, at length 800. At every protein length, Ambient Protein Diffusion's diversity is equal to

its designability, indicating that every designable protein is unique. This is not the case for Proteína, where diversity scores consistently fall below designability, regardless of protein length.

Taken together, these results demonstrate the impact of ambient diffusion on backbone-based generative models and highlight the strength of Genie2's equivariant architecture. Our 17M parameter model trained on approximately 290K AlphaFold structures significantly outperforms a 200M-parameter transformer model trained on roughly 780K proteins. Our results show that smaller, more efficient models can surpass larger transformer baselines in both structural diversity and designability.

**Comparisons on unconditional generation of shorter proteins.** In this experiment, we evaluate the model on the unconditional generation of shorter proteins in Figure 4. The Ambient Protein Diffusion model used in this experiment was trained on a dataset filtered with a TM-Align threshold of 0.4 (as opposed to 0.5), resulting in a training set of approximately 90K cluster representative proteins.

Following the Genie2 protocol, we generate 5 structures for each sequence length from 50 to 256 residues, yielding a total of 1,035 structures. The generated structures are evaluated for both designability and diversity. In line with prior work, we sweep the noise scale $\gamma$ to explore the tradeoff between designability and diversity. Ambient Protein Diffusion outperforms previous methods on both metrics, establishing a new Pareto frontier that achieves superior performance compared to all existing models, including Proteina.

While it is well known that protein pairs with TM-scores above 0.5 typically share the same fold, and those below 0.5 generally do not, we find that the trade-off between designability and diversity is sensitive to the underlying structural heterogeneity of the dataset. Notably, clustering with a TM-align threshold of 0.4, which corresponds to less than a 1% chance of shared global topology, slightly outperforms the 0.5 threshold, which reflects a ~38% probability of topological similarity [43].

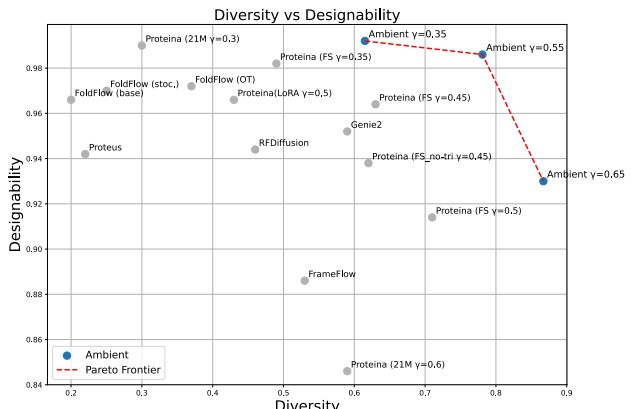

Figure 4: **Designability - diversity trade-off for short protein generation** (up to 256 residues). Ambient dominates completely the Pareto frontier between designability and diversity, while using a $12.88\times$ smaller model. We further do so without using any higher-order sampler or (auto-) guidance method.

**Ablating the Significance of Ambient Diffusion.** We validate the effectiveness of Ambient Diffusion in Figure 5. For our ablation, we perform the identical three stages of training on a model variant that performs standard diffusion training as opposed to Ambient Diffusion. The Genie2 baseline is taken from the Proteína paper. We find that while our two models perform similarly on proteins of 300 residues, the designability and diversity of a vanilla diffusion model diminishes much more significantly with longer proteins. For proteins with 800 residues, the number of designable clusters drops from 68% to 25%.

**Comparisons on Motif Scaffolding.** We additionally compare our method to prior work on motif scaffolding in Figure 6, with full results provided in the supplement. Our evaluation follows the Genie2 benchmark, which comprises 24 single-motif and 6 multi-motif design tasks [27, 40]. For each task, we generate 1,000 scaffold samples using a noise scale of $\gamma = 0.45$. A design is considered successful if it (1) satisfies Genie2's motif designability criteria and (2) preserves the motif with an RMSD below 1Å. Among successful designs, a scaffold is counts as unique if its TM-score is at most 0.6 when compared to any other successful scaffold. A task is considered solved if at least one successful scaffold is generated.

With $\gamma = 0.45$, Ambient Protein Diffusion generates 1,923 unique successful scaffolds for single-motif tasks, a significant improvement over Genie2's 1,445 [27] and performs comparably to Proteína's 2,094 [19]. Notably, all methods solve a similar number of motifs – RFDiffusion solves 22 of

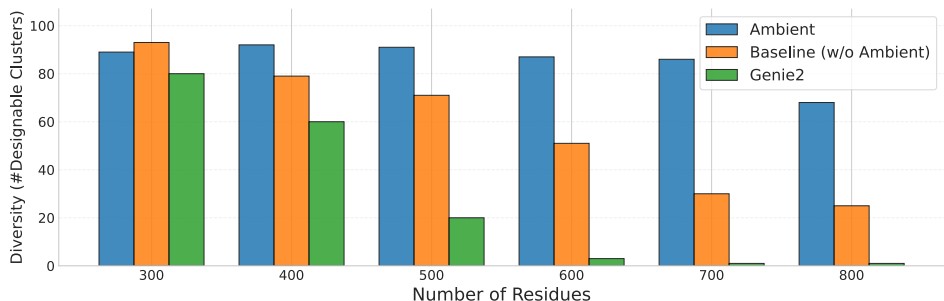

Figure 5: **Effect of ambient diffusion on long protein generation.** We sample protein sequences ranging from 300 to 800 residues. The baseline model (without Ambient Diffusion) shares the same architecture and training dataset, differing only in the diffusion loss and sampling procedure. At a sequence length of 300, the baseline yields four additional designable clusters. However, Ambient Protein Diffusion consistently outperforms both the baseline and Genie2, with increasingly significant improvements as sequence length grows.

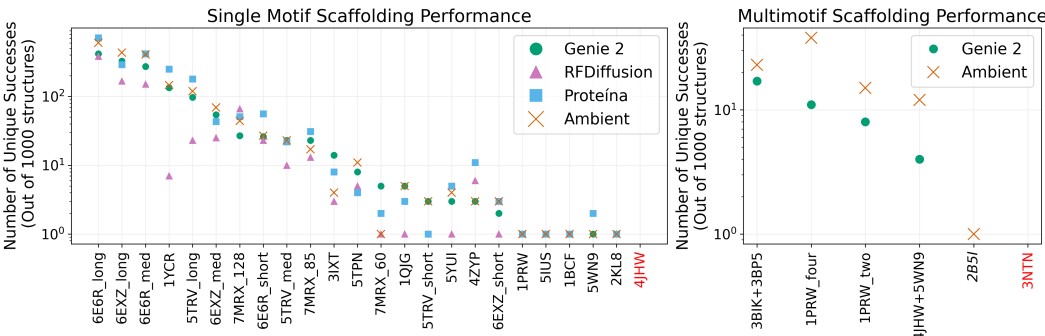

Figure 6: **Performance on Motif Scaffolding Tasks.** We compare Ambient Protein Diffusion to state-of-the-art models for motif scaffolding. The graphs show the number of unique successful scaffolds generated for each single- and multi-motif task. No model produced successful scaffolds for 4JHW and 3NTN. Only Ambient Protein Diffusion produced a valid solution for multi-motif scaffolding of *2B5I*.

the 24 tasks, while Ambient Protein Diffusion, Genie2, and Proteína each solve the same 23 tasks. For multi-motif scaffolding, Ambient Protein Diffusion generates 89 unique successful structures across 5 of the 6 benchmark problems, outperforming Genie2, which produces 40 and solves 4. Ambient Protein Diffusion performs particularly well on the 1PRW_four motif (38 vs. 11 successful structures) in which a scaffold is generated surrounding a calcium binding motif [38]. Overall, Ambient Protein Diffusion outperforms existing methods such as Genie2 and RFDiffusion on single-motif tasks and matches the performance of a Proteína model optimized specifically for motif-scaffolding.

## 5  Conclusion

We introduced *Ambient Protein Diffusion*, a diffusion-based model for protein structure generation that leverages low-confidence AlphaFold structures as a source of noisy training data. Ambient Protein Diffusion enables the generation of long protein structures with unprecedented levels of designability and diversity. Diversity increases as it can use low-confidence Alphafold structures that are typically discarded and designability increases as we separate the pristine quality proteins structures from the medium quality AlphaFold predictions. Ambient Protein Diffusion represents a foundational step toward robust de novo protein design at more natural, biologically relevant lengths. Despite progress, Ambient Protein Diffusion still favors generating alpha-helical structures and developing techniques that address this bias is a crucial direction for future work.

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

# Ambient 🛡 Proteins
## Training Diffusion Models on Noisy Structures
## Supplement

**Anonymous Author(s)**
Affiliation
Address
`email`

## A   Evaluation Metrics

Evaluation of a protein generative model is challenging and there have been a few metrics that have been proposed. In what follows, we explain standard metrics in the protein-generative modeling literature that we will use in our Experimental Results section. Our experiments report using Proteína's definitions of the metrics when possible.

**Designability** (also referred to as refoldability) assesses the structural plausibility of generated proteins. Given a generated backbone, ProteinMPNN [1] generates eight plausible amino acid sequences for that backbone. ESMFold then folds each sequence and the resulting eight structures are compared to the original backbone. The self-consistency RMSD (scRMSD) is defined as the smallest root mean squared deviation between the generated backbone and each of the eight refolded structures. A backbone is considered *designable* if scRMSD $< 2$ Å and designability is defined as the percentage of generated backbones that meet this criterion.

**Diversity** quantifies the structural variability among the generated proteins. Designable backbones are clustered using Foldseek with a TM-score threshold of 0.5. Diversity is then defined as:

$$\text{Diversity} = \frac{\text{Number of Designable Clusters}}{\text{Number of Designable Samples}}.$$

This metric reflects the proportion of structurally distinct (i.e., non-redundant) designable backbones among all designable samples.

## B   Secondary Structure conditioning

Previous work [2] has explored conditioning protein structure generation on CATH labels, a form of hierarchical classification derived from the orientation and spatial organization of protein secondary structures [5]. In this setting, every residue in a protein sequence is typically assigned the same CATH label. In contrast, we propose a more fine-grained approach. Rather than relying on the manually curated and coarse-grained CATH classification, we condition our model directly on secondary structure annotations at the residue level. Each residue is assigned a label corresponding to its local secondary structure (e.g., helix, strand, coil), allowing the model to leverage localized structural context during generation.

We train a variant of our model with partial conditioning, in which the model is conditioned on the secondary structure sequence, without introducing any additional modifications to the input or architecture. We show designable samples conditioned on the secondary structure extracted from real

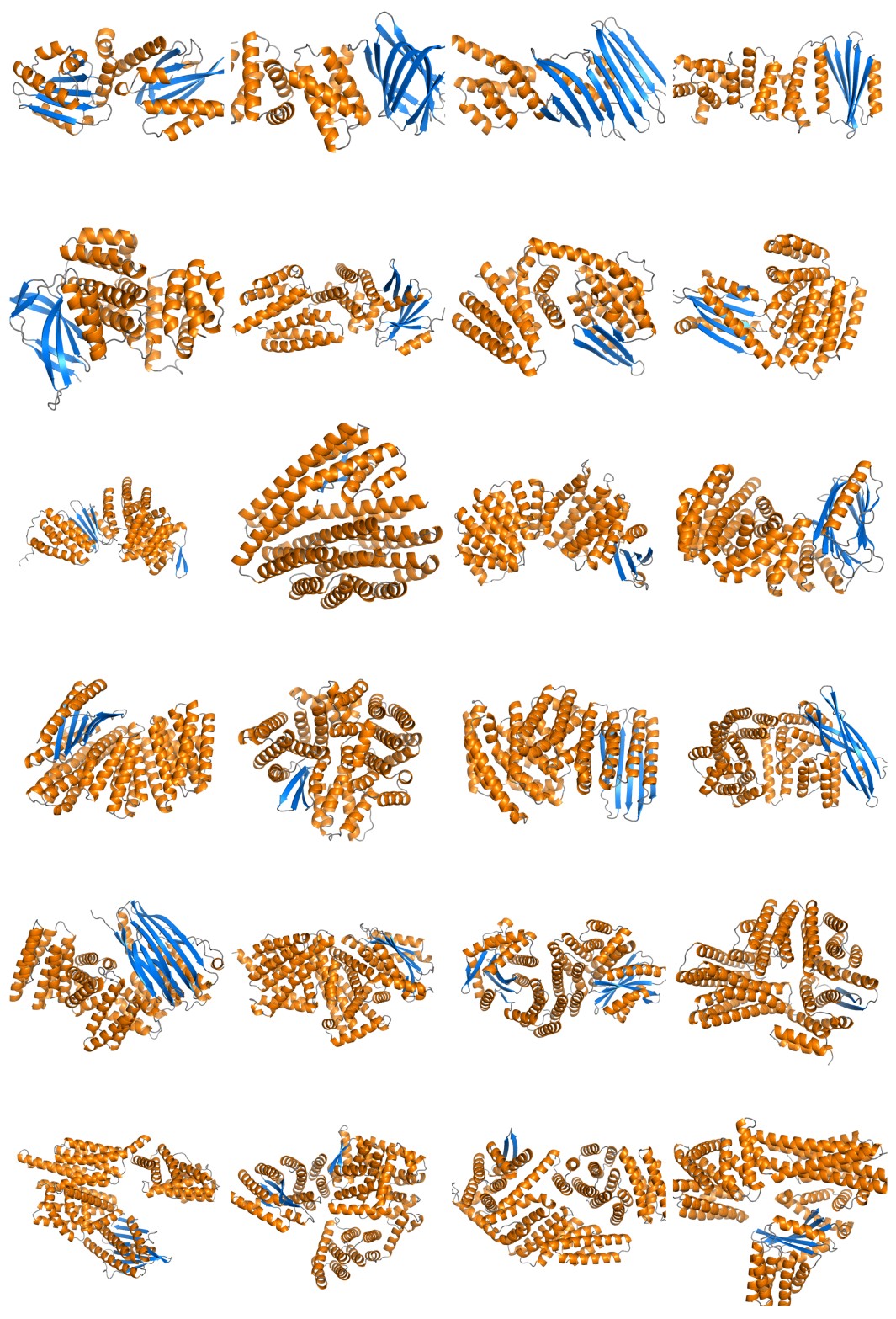

Figure 1: **Qualitative visualizations of unconditional generations.** Our model is capable of producing diverse, multi-domain long proteins.

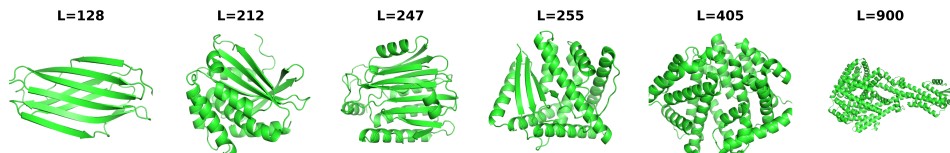

| L=128 | L=212 | L=247 | L=255 | L=405 | L=900 |

Figure 2: **Secondary Structure Conditioned Samples**. We generate proteins using a model variant trained with secondary structure conditioning. To guide generation, we extract secondary structure strings from existing proteins and use this coarse-grained structural representation as input. This conditioning enables the model to produce diverse and designable protein structures.

proteins in Figure 2. These results demonstrate that, even with coarse-grained secondary structure conditioning, our model can generate long, diverse proteins exhibiting a wide range of folds.

## C   Model and Training Hyperparameters

Table 1 includes a more thorough list of the hyperparameters used for our experiments.

## D   Full Motif Scaffolding Results

Table 2 and table 3 represent the numbers of unique successful scaffolds generated by Genie2 [3], RFDiffusion [6], Proteína [2] and *Ambient Protein Diffusion* for each motif in the benchmark dataset in Genie2.

## E   Full Training Algorithm and Implementation Details

### E.1   Additional Implentation Details

**Loss buffer.** The loss rescaling introduced in the main paper ensures balanced weighting across noise levels. At the same time, it also introduces a potential instability: the loss explodes as $\sigma(t)$ approaches $\sigma(t_i)$. To mitigate this instability, we define a buffer zone around each protein's assigned noise level. Specifically, given a protein's assigned noise level $t_i$, it is only used during training at timesteps $t + \tau$, where $\tau$ is a buffer hyperparameter that controls the exclusion margin. This constraint prevents the model from encountering degenerate gradient behavior near the rescaling boundaries and is only applied to medium and low confidence structures (pLDDT < 90). We underline that is similar to how in normal diffusion there is a buffer time zone around $t = 0$ that is never sampled.

**Ambient in high-noise regime.** As explained in the main paper, each protein is only used for a subset of diffusion times according to its average pLDDT value. The proteins that have super high PLDDT ($> 90$) are considered clean data and can be used with the normal training objective. However, as found in [4], using the Ambient training objective for high-noise might be useful even if clean data is available. Intuitively, this objective prevents memorization and promotes diversity in the outputs. We ablated this design choice, and we found a slight increase in diversity for the same designability by using this. Hence, we used this tool from [4] for all our Ambient Protein Diffusion trainings.

### E.2   Algorithm

We provide the full algorithm in Algorithm 1. We commit to open-sourcing our code and models to facilitate the broader adoption of our method from the community.

| Hyperparameter | Genie2 | *Ambient* (Stage 1) | Stage 2 | Stage 3 |
|---|---|---|---|---|
| **Diffusion** | | | | |
| Number of timesteps | 1,000 | - | - | - |
| Noise schedule | Cosine | - | - | - |
| Ambient walls | - | (600,900) | (600,900) | (600,900) |
| **Model Architecture** | | | | |
| Single feature dimension | 384 | - | - | - |
| Pair feature dimension | 128 | - | - | - |
| Pair transform layers | 5 | 8 | 8 | 8 |
| Triangle dropout | 0.25 | - | - | - |
| Structure layers | 8 | - | - | - |
| **Training** | | | | |
| Optimizer | AdamW | - | - | - |
| Number of training proteins | 586k | 196k | 269k | 291k |
| Number epochs | 40 | 200 | 50 | 20 |
| Warmup iterations | 10,000 | 1,000 | 500 | 100 |
| Total batch size | 384 | 384 | 96 | 48 |
| Learning rate | $1.0 \times 10^{-4}$ | $1.0 \times 10^{-4}$ | $1.0 \times 10^{-5}$ | $1.0 \times 10^{-5}$ |
| Weight decay | 0.05 | - | - | - |
| Minimum protein length | 20 | 20 | 50 | 50 |
| Maximum protein length | 256 | 256 | 512 | 768 |
| Minimum mean pLDDT | 80 | 70 | 70 | 70 |
| **Compute Resources** | | | | |
| Number of GPUs | 48 | 48 | 48 | 48 |
| Training time | 18 hr | 18hr | 48hr | 48hr |

Table 1: **Hyperparameters of the diffusion protein model.** Dashes (-) indicate that the value is the same as the previous column. The Ambient walls correspond to the assigned diffusion times based on the protein's PLDDT (times are from 1 to 1000). Proteins with PLDDT $> 90$ are used everywhere. Proteins with PLDDT $> 80$ are used for times in $[600, 1000]$ and proteins with PLDDT $> 70$ are used for times in $[900, 1000]$. We underline that these hyperparameters were not particularly optimized, and even more benefits might be observed by properly tuning these values.

| Motif Name | Genie 2 | RFDiffusion | Proteína | *Ambient Protein Diffusion* |
|---|---|---|---|---|
| **6E6R_long** | 415 | 381 | **713** | 601 |
| **6EXZ_long** | 326 | 167 | 290 | **432** |
| **6E6R_med** | 272 | 151 | **417** | 406 |
| **1YCR** | 134 | 7 | **249** | 146 |
| **5TRV_long** | 97 | 23 | **179** | 119 |
| **6EXZ_med** | 54 | 25 | 43 | **69** |
| **7MRX_128** | 27 | 66 | **51** | 44 |
| **6E6R_short** | 26 | 23 | **56** | 27 |
| **5TRV_med** | **23** | 10 | 22 | **23** |
| **7MRX_85** | 23 | 13 | **31** | 17 |
| **3IXT** | **14** | 3 | 8 | 4 |
| **5TPN** | 8 | 5 | 4 | **11** |
| **7MRX_60** | **5** | 1 | 2 | 1 |
| **1QJG** | **5** | 1 | 3 | **5** |
| **5TRV_short** | **3** | 1 | 1 | **3** |
| **5YUI** | 3 | 1 | **5** | 4 |
| **4ZYP** | 3 | 6 | **11** | 3 |
| **6EXZ_short** | 2 | 1 | **3** | **3** |
| **1PRW** | **1** | **1** | **1** | **1** |
| **5IUS** | **1** | **1** | **1** | **1** |
| **1BCF** | **1** | **1** | 1 | **1** |
| **5WN9** | 1 | 0 | **2** | 1 |
| **2KL8** | **1** | **1** | 1 | 1 |
| **4JHW** | 0 | 0 | 0 | 0 |
| **Total** | 1445 | 889 | **2094** | 1923 |

Table 2: **Detailed single motif scaffolding results.** Ambient Protein Diffusion achieves superior results to Genie 2 and RFDiffusion and performs on par with Proteina. Crucially, our model achieves these results zero-shot, i.e., unlike Proteina, it is not optimized for motif scaffolding and still achieves comparable performance while being an order of magnitude smaller.

| Motif Name | Genie 2 | *Ambient Protein Diffusion* |
|---|---|---|
| **3BIK+3BP5** | 17 | **23** |
| **1PRW_four** | 11 | **38** |
| **1PRW_two** | 8 | **15** |
| **4JHW+5WN9** | 4 | **12** |
| **2B5I** | 0 | **1** |
| **3NTN** | 0 | 0 |
| **Total** | 40 | **89** |

Table 3: **Multi-motif scaffolding results.** Ambient Protein Diffusion achieves consistently superior results to the predecessor Genie-2 model, despite using the same architecture, i.e. the benefit comes from better use of the data. The motif 2B5I is only solved by Ambient Protein Diffusion.

---

**Algorithm 1** Ambient Protein Diffusion: Training Algorithm.

---

**Require:** untrained network $h_\theta$, dataset $\mathcal{D} = \{(x_0^{(i)}, \text{pLDDT}^{(i)})\}_{i=1}^N$, pLDDT to diffusion time mapping function $f : [0, 100] \mapsto \mathbb{R}^+$, noise scheduling $\sigma(t)$, batch size $B$, diffusion time $T$, buffer $\tau$.

1: $\tilde{\mathcal{D}} \leftarrow \left\{ \left(x_0^{(i)} + f(\text{pLDDT}^{(i)})\epsilon^{(i)}, f(\text{pLDDT}^{(i)})\right) | (x_0^{(i)}, \text{pLDDT}^{(i)}) \in \mathcal{D}, \epsilon^{(i)} \sim \mathcal{N}(0, I_d) \right\}$ ▷ Noise each point in the training set according to its pLDDT and get (noisy, noise level) pairs.

2: **while** not converged **do**

3:    $t_s^{(1)}, ..., t_s^{(B)} \leftarrow$ Sample uniformly B times in $[0, T]$ ▷ Sample diffusion times for this batch.

4:    $\tilde{\mathcal{D}}_p \leftarrow \text{shuffle}(\tilde{\mathcal{D}})$     ▷ Shuffle dataset.

5:    $\text{loss} \leftarrow 0$     ▷ Initialize loss.

6:    $\text{pos} \leftarrow 0$     ▷ Initialize index at shuffled dataset.

7:    **for** $i \in [1, B]$ **do**

8:       **while** True **do**     ▷ find the first eligible point

9:          $y, t_y \leftarrow \tilde{\mathcal{D}}_p[\text{pos}]$

10:          **if** $t_y \geq t_s^i + \tau$ **then**

11:             break

12:          **else**

13:             $\text{pos} \leftarrow \text{pos} + 1$     ▷ Move to the next point in the dataset.

14:          **end if**

15:       **end while**

16:       $\epsilon \sim \mathcal{N}(0, I)$     ▷ Sample noise.

17:       $t \leftarrow t_s^{(i)}$     ▷ Time to be used in this training update.

18:       $t_i \leftarrow t_y$     ▷ Assigned time based on the PLDDT value

19:       $x_{t_i} \leftarrow y$     ▷ Noised point to the assigned time.

20:       $x_t \leftarrow x_{t_i} + \sqrt{\sigma^2(t) - \sigma^2(t_i)}\epsilon$     ▷ Add additional noise.

21:       $\alpha(t, t_i) \leftarrow \frac{\sigma^2(t) - \sigma^2(t_i)}{\sigma^2(t)}$.

22:       $w(t, t_i) \leftarrow \frac{\sigma^4(t)}{(\sigma^2(t) - \sigma^2(t_i))^2}$.     ▷ Loss reweighting.

23:       $\text{loss} \leftarrow \text{loss} + w(t, t_i) \left\|\alpha(t, t_i)h_\theta(x_t, t) + (1 - \alpha(t, t_i))x_t - x_{t_i}\right\|^2$    ▷ Ambient loss

24:    **end for**

25:    $\text{loss} \leftarrow \frac{\text{loss}}{B}$     ▷ Compute average loss.

26:    $\theta \leftarrow \theta - \eta\nabla_\theta\text{loss}$     ▷ Update network parameters via backpropagation.

27: **end while**

---

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
