# OpenReview forum: "Ambient Proteins: Training Diffusion Models on Low Quality Structures"
_NeurIPS.cc/2025/Workshop/Reliable_ML — NeurIPS 2025 - Reliable ML Workshop_

### Official Review · Reviewer_jfqf · 2025-09-19
**Peer review of Ambient Proteins: Training Diffusion Models on Low Quality Structures**

**Rating:** 6
**Confidence:** 2

**Review:**

Summary:
The paper introduces a new diffusion-based method for protein structure generation that is able to salvage low-confidence samples from the large AlphaFold database. This is done by training a transformer model derived from Genie2 with some slight architecture changes. The resulting model is comparable yet smaller in size to current alternatives such as Proteína. Results show that the proposed method is able to increase geometric diversity in generated protein structures by taking advantage of the low-quality samples. Additionally, it is shown that their model outperforms in the designability-diversity tradeoff quite significantly.

Strengths:
The authors were able to design a diffusion model that outperforms similar models with less parameters. This is a clever approach that found a way to take advantage of available, but previously discarded samples due to their low confidence nature. This works by leveraging the ambient diffusion that is designed for problems like protein generation that are difficult to find quality examples.

Weaknesses:
The authors mention in the conclusion that the model contains bias for alpha-helical structures, and this is a direction for future work. For audiences' that are not involved in the specific domain, it is not obvious what results were presented that this applies to. If this is a current limitation, it is important for the general audience to clearly understand what this bias is.

Suggestions:
More explicit discussion of the bias towards specific structures is necessary to speak to a general audience. The mention of secondary structures in the supplement introduces the concept, but this bias towards alpha-helical structures is not highlighted in the main text. Since the topic area is related to reliable machine learning, there should be more focus on this limitation in the results or in a limitations section.

---

### Official Review · Reviewer_sHKN · 2025-09-20
**This paper proposes a new method for deriving new protein structures.**

**Rating:** 5
**Confidence:** 3

**Review:**

Strenghts:
1) Improved designability, diversity than the previous state of the art.
2)New dataset for geometric diversity.


Weaknesses/Questions:

1) Do you have any experiments on Generative model training other than Protein Diffusion to evaluate the performance of your method.
2) In lines 142-144, do you have any proof or reference for this claim? (one can prove that the conditional expectation...)
3)From a presentation perspective, the pseudocode of the algorithm would be nice to be included in the main text.

I believe that the authors have done a great work, but I don't think it fits this workshop. I would like to see the evaluation of their method to a greater range of Machine Learning Application and , maybe, some mathematical results about its performance.